# Is a Caption Worth a Thousand Images?
# A Study on Representation Learning

**Shibani Santurkar, Yann Dubois, Rohan Taori, Percy Liang & Tatsunori Hashimoto**
Stanford University
`{shibani,yanndubs,rtaori,pliang,thashim}@stanford.edu`

## Abstract

The development of CLIP (Radford et al., 2021) has sparked a debate on whether adding language supervision can yield vision models with more transferable representations than traditional image-only methods. Our work studies this question through a *carefully controlled* comparison of two approaches, in terms of their ability to learn representations that generalize to downstream classification tasks. We find that when the pre-training data meets certain criteria—it is sufficiently large and contains descriptive captions with low variability—-image-only methods *do not* match CLIP's performance even when they are trained with more image data. However, contrary to what one might expect, there are practical settings in which these criteria are not met, wherein added supervision through captions is actually *detrimental*. Motivated by our findings, we devise simple data and algorithmic interventions to improve the transfer performance of CLIP-style models.

## 1 Introduction

Image-based contrastive learning approaches have shown promise in building models that generalize beyond the data distributions they are trained on (Wu et al., 2018; He et al., 2020; Chen et al., 2020a; Caron et al., 2020; Chen et al., 2020b; Caron et al., 2021). By leveraging large (unlabelled) data sources via self-supervised training, these models learn representations that transfer to diverse image classification tasks—more so than their supervised counterparts (Ericsson et al., 2021).

Recently, Radford et al. (2021) showed that a different approach—contrastive learning with language supervision—can yield models (CLIP) with remarkable transfer capabilities. This development has garnered significant interest in the vision and natural language processing communities alike, leading to a debate on the utility of multi-modality in visual representation learning (Zhai et al., 2022; Devillers et al., 2021; Fang et al., 2022). Our work focuses on a specific question within this debate:

*Does added language supervision lead to more transferable visual representations*
*than using images alone?*

It might seem like the answer to this question is obvious. After all, CLIP utilized caption information unavailable to traditional image-based approaches and showed substantial gains over them (Radford et al., 2021). However, CLIP is drastically different from these approaches in many ways, from training data to fine-grained implementation choices, which makes it difficult to isolate the contribution of language supervision (see Section 5). Further, recent studies on CLIP's zero-shot classification and robustness properties cast doubt on whether adding language supervision is always beneficial (Fang et al., 2022). Resolving the aforementioned debate thus requires a carefully controlled comparison of the two approaches in which the *only* difference is the form of supervision.

**Our contributions.** We devise a methodology to assess the utility of language supervision in CLIP[1] from a visual representation learning standpoint. To do so, we recognize that CLIP pre-training and popular image-based methods share the same underlying primitive of contrastive learning. Specifically, Radford et al. (2021)'s approach is strikingly similar to SimCLR (Chen et al., 2020a). The only irreducible difference between them is whether supervision is provided to the

---

[1]We use CLIP to refer to models trained with Radford et al. (2021)'s approach, not their pre-trained model.

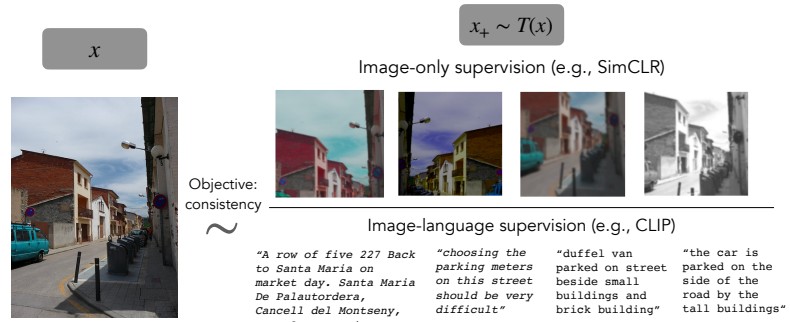

Figure 1: A conceptual view of contrastive image-only and image-language pre-training. Both methods rely on the same self-supervised objective: aligning the representations of positive examples $(x, x_+)$ while distinguishing them from negative ones $(x_n)$. The transformation $T(\cdot)$ used to obtain $x_+ \sim T(x)$ (augmented image or caption) encodes the equivalences the model must satisfy.

model via image augmentations or image-caption matching (see Figure 1)—which is precisely the quantity we want to study. Thus, we can disentangle the effect of language supervision on visual representations by comparing matched versions of SimCLR and CLIP (trained from scratch). Our focus, in particular, is on how well the learned representations transfer to varied image classification tasks. We find that the picture is nuanced and depends on three properties of the pre-training data:

1. When the *scale* of the dataset is sufficiently large, CLIP's visual representations indeed transfer better than their matched image-only SimCLR counterparts. In fact, this gap is not bridged by training SimCLR with more (image) data, suggesting that a caption can be worth more than *any* number of images. However, in the low-data regime, language supervision actually hurts model performance both in and out-of-distribution.

2. The *descriptiveness* (Kreiss et al., 2021) of captions—i.e., the extent to which they refer to what is contained in an image—directly determines how well CLIP models transfer. In fact, we find that a single descriptive image-caption pair (e.g., from COCO (Lin et al., 2014)) is worth five less descriptive, uncurated captions (e.g., from YFCC (Thomee et al., 2016)).

3. The *variability* of captions (e.g. stylistic or lexical) within a dataset can impair CLIP's performance. We find that a modification to standard CLIP training—performing text augmentations by sampling from a pool of captions for each image—can alleviate this drop.

These properties have inter-twined effects on CLIP's performance: e.g., dataset scale can, to some extent, compensate for less-descriptive and/or varied captions. Guided by our findings, we devise simple datasets interventions that can lead to more-transferrable CLIP models: (i) filtering out low-quality captions with a text-based classifier, and (ii) applying data augmentation to captions by paraphrasing them using pre-trained language models.

## 2 AN APPLES-TO-APPLES COMPARISON

Prior works have studied image-only and image-language pre-training methods in isolation (Wu et al., 2018; He et al., 2020; Chen et al., 2020a; Caron et al., 2020; Chen et al., 2020b;b; Chen & He, 2021; Caron et al., 2021; Radford et al., 2021) and side-by-side (Desai & Johnson, 2021; Devillers et al., 2021; Fang et al., 2022). Yet, they provide incomplete (and often contradictory) answers to our motivating question of the value of language supervision relative to using images alone (Section 5). Crucially, this is due to various confounders such as: (i) bespoke algorithmic optimizations within the two methods, and (ii) differing pre-training datasets. In this section, we outline a series of steps that we take to mitigate these confounders and compare the two methods on equal footing.

### 2.1 FINDING COMMON GROUND

Our approach for studying the value of language supervision is guided by the following insight: CLIP pre-training is strikingly similar to the popular image-only SimCLR method (Chen et al.,

2020a)[2]. Both methods rely on the same algorithmic primitive of *contrastive learning*, which we illustrate in Figure 1. Specifically, the (CLIP/SimCLR) model is trained cross-entropy based objective, which for a given pair $(x, x_+)$ of positive examples with with associated negatives $\mathcal{N}$ is:

$$\ell = -\log \frac{\exp(\text{sim}(z, z_+)/\tau)}{\sum_{n \in \mathcal{N} \cup \{z_+\}} \exp(\text{sim}(z, z_n)/\tau)} \ , \ \text{ where } z = g(\phi(x)) \text{ and } z_{+/n} = g'(\phi'(x_{+/n})), \quad (1)$$

sim is cosine similarity, $\phi/\phi'$ are encoders, and $g/g'$ are projection heads. Positive examples $x_+$ are obtained through a transformation of the image $x$, i.e., $x_+ \sim T(x)$—such as image augmentations (e.g., rotations or crops) in SimCLR and captions in CLIP. Observe that this difference in $T(\cdot)$ between CLIP and SimCLR corresponds exactly to whether the model is trained with language, which is the quantity we want to study. Thus, to isolate the role of added language supervision, we can compare the downstream performance of matched CLIP and SimCLR models. To this end, we must take some steps to make their implementatations consistent:

- *Datasets:* Typically, CLIP and SimCLR are trained on different datasets, as the former requires image-caption pairs, while the latter can leverage any image data. To control for the effect of the data distribution, we pre-train both models *from scratch* on the same data.
- *Architecture:* We use the ResNet-50 (He et al., 2016) architecture as the image encoder for both methods, and a Transformer (Vaswani et al., 2017) as the text encoder in CLIP. We also extensively tune hyperparameters for both methods (Appendix A.3).
- *Augmentations:* Both methods apply data augmentations to the image $x$ itself at each training step. However, the augmentations used in SimCLR (`resize`, `crop`, `flip`, `jitter`, `blur`, `grayscale`) are far more sophisticated than those in CLIP (`resize` and `crop`). We remove this confounder by using SimCLR augmentations unless otherwise specified.
- *Transformation stochasticity:* The two methods differ in how they obtain $x_+$, not just due to the choice of $T(x)$ but also the generative process itself. In SimCLR , $x_+$ is a new random draw from $T(x)$ in every batch, while for CLIP, it is a single fixed caption. Perfectly matching them requires training CLIP by sampling a fresh caption $x_+$ for each image at each iteration. We will refer to this stochastic version of CLIP as CLIP$_\text{S}$.

**Mismatches.** Despite our efforts to match CLIP with SimCLR, some inconsistencies remain–partly due to their differing input modalities. In particular, CLIP (and CLIP$_\text{S}$):

(i) Processes $T(x)$ using a text transformer rather than SimCLR's ResNet-50.

(ii) Does not share weights between the encoders processing $x$ and $T(x)$ because they correspond to different modalities, unlike SimCLR.

(iii) Uses a linear projection head $g/g'$ instead of SimCLR's MLP, which we allow as Radford et al. (2021) showed that this choice does not affect CLIP's performance.

(iv) Only uses other examples in the batch from the *same* modality as negatives. Thus CLIP has half the number of negatives compared to SimCLR, which also uses transformed versions of other examples in the batch (i.e. both $\hat{x}$ and $\hat{x}_+$) as negatives.

We now assess how the representations learned by our matched CLIP and SimCLR models compare. In particular, we measure how well their representations transfer to the downstream tasks from Kornblith et al. (2019). Akin to (Radford et al., 2021), we focus on the *fixed-feature* setting, where we freeze the weights of a given model and then train a linear probe using task data (see Appendix A).

## 2.2 A CASE STUDY

We begin by comparing CLIP and SimCLR models trained on the MS-COCO dataset (Lin et al., 2014) (henceforth referred to as COCO), which contains ∼120K images with multi-object labels. Each image has five human-provided captions, collected post-hoc by Chen et al. (2015) using Mechanical Turk. Annotators were given detailed instructions on how to caption an image such as to describe *only* the important parts of the image and not to use proper names. We use COCO as our

---

[2]Other image-based methods (He et al., 2020; Chen et al., 2020b; Chen & He, 2021; Caron et al., 2021) have optimizations that are not present in CLIP.

Table 1: Linear probe accuracy for COCO pre-trained models in-distribution and on the transfer suite from Kornblith et al. (2019) ($\mu_{Tx}$ denotes average transfer accuracy).

| | COCO | Aircraft | Birdsnap | Ctech101 | Ctech256 | Cars | CIFAR10 | CIFAR100 | DTD | Flowers | Food-101 | Pets | SUN937 | $\mu_{Tx}$ |
|---|---|---|---|---|---|---|---|---|---|---|---|---|---|---|
| Supervised | **90.6** | 31.6 | 11.8 | 65.8 | 53.7 | 21.7 | 74.8 | 46.7 | 55.9 | 63.4 | 47.1 | 45.9 | 44.5 | $47.2 \pm 0.2$ |
| SimCLR | 89.0 | 40.6 | 18.5 | 71.5 | 58.6 | 31.5 | 82.1 | 57.3 | 61.7 | 77.4 | 58.7 | 57.3 | 51.9 | $56.0 \pm 0.2$ |
| CLIP | 88.4 | 41.4 | 17.6 | 73.2 | 60.4 | 35.8 | 83.6 | 60.8 | 65.7 | 80.5 | 60.9 | 57.0 | 50.8 | $57.5 \pm 0.1$ |
| CLIP$_S$ (5) | 89.8 | **46.4** | **20.0** | **78.4** | **65.6** | **41.5** | **84.6** | **62.5** | **66.7** | **83.9** | **65.3** | **61.2** | **54.9** | $\mathbf{61.3 \pm 0.2}$ |

starting point for two reasons. First, we can assess the utility of language supervision in the ideal setting where the captions are of fairly high quality due to the careful curation process. Second, we can approximate CLIP$_S$[3] by sampling from the available set of five captions per image.

**Captions (often) help on COCO.** In Table 1, we compare various COCO pre-trained models (supervised, SimCLR, CLIP/CLIP$_S$) in terms of the accuracy of a linear probe on: (i) COCO classification (in distribution), and (ii) transfer tasks. Note that to contrast image-only and image-language supervision, the "right" comparison is between SimCLR and CLIP$_S$: they are matched (to the best of our abilities) in terms of dataset, architecture, augmentations and stochasticity. We find that:

- As expected, the supervised model outperforms self-supervised ones in distribution, but the trend flips when we consider transfer tasks.

- Our matched stochastic variant of CLIP, CLIP$_S$, outperforms SimCLR both in- ($\sim 1\%$) and out-of-distribution ($\sim 6\%$).

- Vanilla CLIP (which does not contain stochastic transformations like SimCLR/CLIP$_S$) still performs better than SimCLR on downstream transfer tasks: albeit by a smaller margin ($\sim 2\%$). Further, CLIP is worse in-distribution than SimCLR. In Section 3.3, we take a closer look at exactly *why* stochasticity in CLIP has such a drastic impact on performance.

- Finally, matching image augmentations (applied to $x$) turns out to be crucial to assessing the merit of added language supervision. Using typical CLIP augmentations (only `resize` and `crop`) during training lowers CLIP's average transfer accuracy by 10% (Appendix Table 4) and SimCLR's by 50%. This highlights the importance of controlling for potential confounders (Section 2.1) to fairly evaluate the utility of added language supervision.

## 3 THE IMPACT OF PRE-TRAINING DATA

Our analysis of COCO shows that language supervision can be beneficial over using images alone. That being said, the datasets that CLIP is typically trained on differ, both in scale and quality, from COCO. For instance, COCO captions were collected post-hoc under controlled settings, which is markedly different from the automatic scraping procedure used to gather data at scale. Thus, we shift our focus to two frequently-used (Ilharco et al., 2021) CLIP training datasets:

*ConceptualCaptions* (Sharma et al., 2018) (CC) contains $\sim$3.3M images harvested from web, with their ALT-text attributes as captions. The data was filtered for text quality—e.g., well-formed captions that mention at least one object found via the Google Cloud Vision API. Furthermore, all proper nouns in the captions were hypernymized (e.g., "Justin Timberlake" becomes "pop artist").

*Yahoo Flickr Creative Commons* (Thomee et al., 2016) (YFCC): This dataset has $\sim$ 99.2M images from Flickr, along with their posted titles as captions with no post-processing.

---

[3]We henceforth overload notation and use CLIP$_S$ to denote: (i) the idealized stochastic version of CLIP, which samples from infinite captions per image, and (ii) our approximation of it with a finite set of captions.

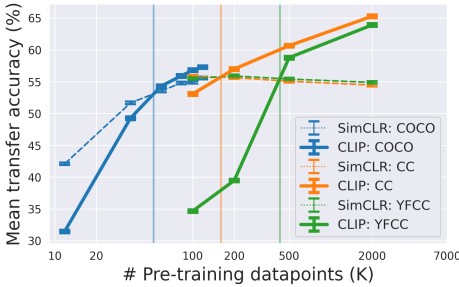 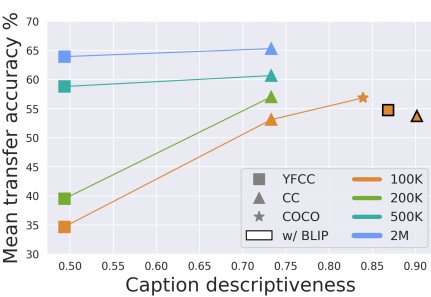

Figure 2: (*left*) Effect of pre-training dataset size on models' performance. While added language supervision consistently improves transfer accuracy in the medium to large data regime over using images alone, it is actually detrimental on small corpora. (Due to compute constraints, we train models for fewer epochs (100 instead of 200) on datasets of size 2M.) (*right*) Relationship between CLIP's transfer performance and the average *descriptiveness* of dataset captions, measured using a pre-trained BLIP caption-scoring model (Li et al., 2022). Additionally, we also visualize the performance of CLIP models trained on 100K YFCC/CC samples using "more descriptive" BLIP-generated captions: cf. the *orange square/triangle outlined in black*.

**Do captions still help?** We start by comparing the transfer performance of CLIP and SimCLR on 100K subsets of COCO/CC/YFCC in Figure 2(*left*). We observe that SimCLR's transfer capabilities do not vary much across pre-training datasets, while CLIP's performance is highly sensitive to them. With 100K samples from CC/YFCC, using CLIP is *worse* than image-only pre-training via SimCLR—unlike what we see for COCO.

**The sensitivity of CLIP to pre-training data.** Inspecting dataset samples (Figure 3) yields a possible explanation for this sensitivity. The three datasets differ not just in scale and image diversity, but also the extent to which captions: (i) *describe* visually salient aspects of the image, and (ii) *vary* across images (e.g., in style and wording). For instance, COCO captions are homogenous and descriptive, while YFCC ones vary and are often complementary to the image. We now study the effect these dataset properties—scale, descriptiveness, and variability—have on CLIP's performance.

## 3.1 SCALE MATTERS

A major appeal of contrastive learning methods is that they can leverage the vast amounts of unlabeled data available on the Internet. Thus, it is natural to ask how different forms of contrastive supervision benefit from added pre-training data. We may expect image-only methods to perform worse for smaller datasets as they are less likely to encounter (augmented) images which are similar. We might further expect image-language models to perform more favorably in this setting since they receive richer supervision. To test whether this is the case, we compare CLIP and SimCLR models trained on datasets of varying sizes: 10-100K samples for COCO, and 100K-2M for CC/YFCC.

Our results in Figure 2(*left*) deviate from our earlier expectations. First, beyond a certain point, SimCLR's transfer performance improves only marginally with additional data. While surprising, similar effects have been noted previously (Tian et al., 2021; Cole et al., 2022), especially when the data is uncurated (e.g., YFCC) (Tian et al., 2021). Second, in the low-data regime (<50K/200K/500K for COCO/CC/YFCC), training with language actually hurts the models' transfer performance. In fact, (data) scale seems to be essential to benefit from language supervision. With sufficient data, CLIP outperforms SimCLR on all three datasets. This gap remains even if we train SimCLR with extra data, indicating that captions can be worth more than *any* number of images.

## 3.2 THE IMPORTANCE OF DESCRIPTIVE CAPTIONS

Prior work in linguistics and accessibility has drawn a distinction between image "descriptions" and "captions" (Berger & Dibb, 2003; Chandler, 2007; Hodosh et al., 2013; Bernardi et al., 2016; van Miltenburg, 2020; Kreiss et al., 2021; Dognin et al., 2022; Hutchinson et al., 2022). In particular, Bernardi et al. (2016) define descriptions as texts that "verbalize what can be seen in the image,

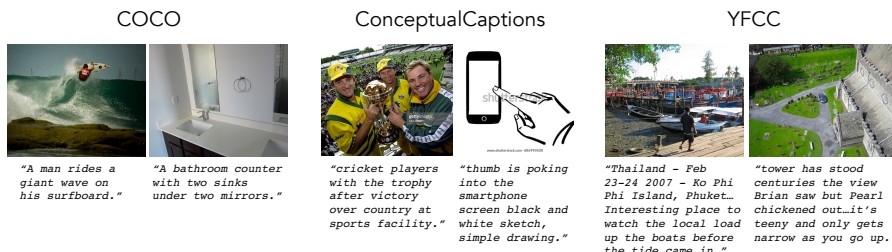

Figure 3: Random samples from the COCO, CC and YFCC datasets (also see Appendix Figure 6).

i.e., they refer to the objects, actions, and attributes depicted, mention the scene type, etc.". In contrast, Panofsky (1939) suggest that a typical caption "provides personal, cultural, or historical context for the image." This line of work suggests that COCO captions are more descriptive due to the decontextualization of the image and strict instructions provided to the annotators during the caption generation process (Kreiss et al., 2021). In contrast, Flickr captions (e.g., in CC/YFCC) tend to contain information that is complementary to the image Alikhani et al. (2020) since people tend not to not restate what can already be observed in the photographs they post (Hodosh et al., 2013).

Now to perform well on downstream classification tasks, we ideally want model representations that encode salient image objects. Recall that in contrastively-trained models, the learned representations are determined by the transformation $T(x)$ (captions for CLIP). This suggests a hypothesis: pre-training CLIP with descriptive captions will yield more transferrable (vision) representations.

To test this, we need to quantify the descriptiveness of a caption. Since doing so precisely is infeasible, we approximate descriptiveness using a pre-trained caption-scoring model. Specifically, we leverage the BLIP model (Li et al., 2022) which has shown state-of-the-art performance on image-based text retrieval. We then measure the average score assigned by BLIP to dataset captions matching their corresponding images—see Figure 2(*right*). As expected based on our earlier subjective assessment as well as prior work (Hodosh et al., 2013; Kreiss et al., 2021), we indeed find that the caption descriptiveness of COCO > CC > YFCC (see Appendix B.1 for a discussion of relevant vs. noisy captions.) Furthermore, we see that the descriptiveness of captions in the pre-training data directly correlates with CLIP's transfer performance. In fact, a CLIP model trained on 100K descriptive image-caption pairs from COCO attains performance comparable to one trained on 2x and 5x more samples from CC and YFCC respectively.

To further corroborate our hypothesis, we train CLIP CC and YFCC with "more descriptive" captions by re-captioning the images using BLIP (Li et al., 2022). Indeed, we find that CLIP trained on 100K CC/YFCC samples with BLIP captions no longer performs worse than its COCO counterpart (see Figure 2(*right*)). This indicates that CLIP's sensitivity to the pre-training corpus is not just an artifact of differing image distributions, but due to the presence (or absence) of descriptive captions.

### 3.3 THE EFFECT OF INTRA-DATASET VARIATIONS IN CAPTIONS

Image captions (Figure 1) seem to vary in how they describe an object (e.g., "duffel van" or "car") and the parts of the image they focus on (e.g., discussing the "street" or "brick"). We now study how these lexical and focus variations in captions CLIP's ability to learn meaningful representations.

**A simple setting.** As a starting point, we investigate this effect on the COCO dataset using *synthetic* captions—constructed using the available multi-object labels—whereby we can precisely control the intra-dataset captions variations. In an attempt to simulate the variations we observe in Figure 1, we design the captions to (not) be: (i) *consistent:* use a fixed term or random synonyms to describe an object across the dataset (lexical variations); and (ii) *complete:* mention all or a random subset of image objects (focus variations). (See Appendix A.6 for details and Appendix Figure 8 for examples.) Surprisingly, we find that a CLIP model trained with complete and consistent synthetic COCO captions *outperforms* a model trained on human-written captions (cf. row 1 in Figure 4(*left*) to row 3 in Table 1). However, dropping these two conditions causes the transfer performance of the model to drop significantly (cf. rows 1, 2, and 4 in Figure 4(*left*)). These findings suggest that variability in dataset captions can have an adverse effect on the resulting CLIP models.

| Consistency | Completeness | Model | COCO | $\mu_{Tx}$ |
|:---:|:---:|:---:|:---:|:---:|
| ✓ | ✓ | CLIP | 88.8 | $59.2 \pm 0.1$ |
| ✓ | ✗ | CLIP | 88.4 | $57.7 \pm 0.2$ |
| ✓ | ✗ | $\text{CLIP}_S$ | 89.1 | $59.3 \pm 0.2$ |
| ✗ | ✗ | CLIP | 88.4 | $56.6 \pm 0.2$ |
| ✗ | ✗ | $\text{CLIP}_S$ | 89.3 | $58.9 \pm 0.2$ |

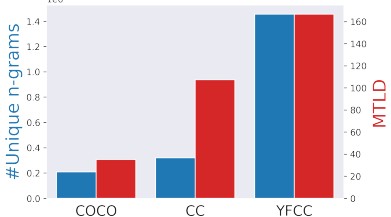

Figure 4: (*left*) CLIP trained on synthetic COCO captions (constructed using multi-object labels): effect of caption variability due to inconsistency (use a single term to describe a given object) and incompleteness (describes all image objects) on transfer performance. (*right*) Quantifying the variability of dataset captions based on the number of unique n-grams (n=1,2,3) (Li et al., 2015; Fung et al., 2020) and the MTLD score (McCarthy & Jarvis, 2010) in practical image-caption datasets.

**The effect of stochasticity.** We now revisit our stochastic CLIP variant, $\text{CLIP}_S$, in this simple setting. Intuitively, we might expect that sampling from a set of diverse captions per image—which cover possibile lexical and stylistic variations—during training might alleviate the adverse effects of caption variability. Indeed, we find that for synthetic COCO captions, $\text{CLIP}_S$ is not as affected by caption inconsistency and/or incompleteness. The $\sim 2\%$ improvement of $\text{CLIP}_S$ over CLIP here mirrors the 3.6% gain seen for human-provided captions (cf. Table 1). These findings suggest that one of the reasons why stochasticity significantly boosts CLIP's performance is its role in (caption) variance reduction. We also find that $\text{CLIP}_S$ transfers 2% better when trained on human-provided captions as opposed to synthetic ones (unlike CLIP). This indicates that human-written captions do contain useful information that is not present in object labels alone. However, extracting this signal is not straightforward, and may require incorporating multiple captions into CLIP training.

**Datasets in practice.** We now attempt to characterize caption variability in real-world datasets. Inspired by prior work in natural language processing and linguistics (Appendix A.8), for a set of dataset captions, we measure: (i) the total number of unique n-grams (N=1-3) (Li et al., 2015; Fung et al., 2020) and (ii) measure of textual lexical diversity (MTLD) (McCarthy & Jarvis, 2010). Along both these axes of variability, we see that COCO < CC < YFCC (Figure 4(*right*)). Thus, aside from the lower descriptiveness of YFCC (and to a lesser extent CC) captions, their variability could be the reason why the resulting CLIP models have worse transfer performance (Figure 2(*left*)). This also explains why scale is essential to benefit from language supervision on CC and YFCC. After all, CLIP would need to be trained on more captions to even encounter the same words twice.

**How many captions are enough?** We saw above that "text data augmentations" via $\text{CLIP}_S$ could reduce the adverse impacts of caption variability. We now analyze how this effect scales with the number of available captions per image on the CC and YFCC datasets. Here, we use the BLIP captioning model to generate multiple captions per image via nucleus sampling (Holtzman et al., 2020). This procedure is intended to serve as a proxy for the manual caption annotation or automated scraping procedures that might be used for data collection in practice. We observe in Figure 5(*left*), that $\text{CLIP}_S$ improves as the number of available captions per image increases (plateauing around 10). However, scaling up the overall number of image-caption pairs appears to be far more effective than incorporating more captions per image (at least those obtained via BLIP) from the perspective of improving transfer performance (see Figure 5(*right*)). Note that the exact trade-offs and their costs are context dependent and vary based on the exact procedure used for caption collection.

## 4 MAKING EXISTING CAPTIONS WORK

So far, we identified three properties of the pre-training data that influence CLIP's transfer performance: (i) scale, (ii) caption descriptiveness, and (ii) caption variability. Our analysis shows that one way to improve CLIP's performance, especially on uncurated data sources, is to simply pre-train with more data. Alternatively, for a fixed data scale, we may be able to obtain better CLIP models if we improve *what* captions describe and *how* they describe an image. We now focus on the latter and put forth simple dataset interventions to improve transfer capabilities of CLIP-style models.

| Method | $N_C$ | Source | CC (100K) | YFCC (100K) |
|--------|-------|--------|-----------|-------------|
| SimCLR | 0 | - | $55.9 \pm 0.2$ | $55.5 \pm 0.2$ |
| CLIP | 1 | Human | $53.1 \pm 0.2$ | $34.7 \pm 0.2$ |
| CLIP | 1 | BLIP | $53.7 \pm 0.2$ | $54.8 \pm 0.2$ |
| $CLIP_S$ | 2 | BLIP | $56.1 \pm 0.2$ | $56.9 \pm 0.2$ |
| $CLIP_S$ | 5 | BLIP | $57.8 \pm 0.2$ | $58.8 \pm 0.2$ |
| $CLIP_S$ | 10 | BLIP | - | $59.1 \pm 0.2$ |

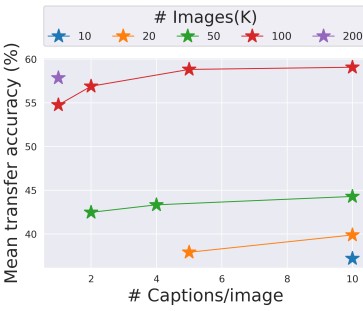

Figure 5: A closer look at $CLIP_S$. (*left*) Sensitivity of its transfer performance to the number of captions per image ($N_C$) used during training. For CC and YFCC, we use the BLIP captioning model to generate multiple diverse captions per image. Note: due to compute constraints, we run CC pre-training for a subset of $N_C$ values. (*right*) Performance trade-offs between pre-training using more image-caption pairs vs. more captions per image on the YFCC dataset.

**Data pre-processing:** Given the importance of caption descriptiveness, we might consider pre-processing scraped data to select for samples with this property. The CC data collection procedure (Sharma et al., 2018) partially demonstrates the effectiveness of this approach, as pre-training CLIP on CC samples leads to better transfer performance than a comparable number of "raw" YFCC ones. However, due to its reliance on the Google Vision API, this procedure can be quite expensive, with costs scaling with the size of the scraped data. Recent works have taken a different approach, using pre-trained image-language models (like CLIP) to filter data (Schuhmann et al., 2021). However, since we are interested in building such models in the first place, we avoid taking this route.

Instead, we focus on understanding how far we can get by simply discarding low quality captions, agnostic to the images. We take inspiration from the filtering pipelines used to build large language models (Brown et al., 2020). Here, raw Internet data is cleaned by selecting samples that are "similar" to known high-quality datasets (e.g., Wikipedia). Taking a similar approach, we train a linear classifier on a bag-of-n-grams sentence embeddings (Joulin et al., 2017) to distinguish validation set CC/YFCC captions from COCO ones. This classifier is then used to filter CC/YFCC, only retaining samples that are predicted as being COCO-like. This simple procedure does end up selecting for captions that are more focused on objects and their descriptions, as opposed to describing contextual properties such as dates or urls—see Appendix A.9. For a given pre-training data budget, we see moderate gains ($\sim 2\%$) from using this heuristic to filter datasets—see Table 2 (*left*).

**Mitigating caption variability:** As we saw in Section 3.3, models trained with $CLIP_S$ are less impacted by caption variability. However, typical image-captioning datasets (such as CC and YFCC) only have one caption per image. We thus devise a methodology to augment these captions by leveraging recent open-source large language models (Wang & Komatsuzaki, 2021). Concretely, we provide GPT-J with $4$ (caption, paraphrase) pairs as *in-context* (Brown et al., 2020) examples. We then prompt it to paraphrase a given target caption. By sampling from GPT-J, we can obtain multiple (in our case, five) paraphrases for every such caption (examples in Appendix Figure 10). In Table 2 (*right*), we see that feeding these captions into $CLIP_S$ results in a considerable performance boost over CLIP (trained with a single caption/image). For instance, for COCO, $CLIP_S$ trained on our generated captions bridges more than half of the performance gap between vanilla CLIP and $CLIP_S$ trained with five human-provided captions.

## 5 RELATED WORK

**Representation learning.** Building models with *general* representations that transfer to downstream tasks has been a long-standing goal in ML (Donahue et al., 2014; Razavian et al., 2014; Chatfield et al., 2014; Agrawal et al., 2014; Yosinski et al., 2014). Our work is in line with prior studies aimed at characterizing the effect of design choices made during training (Azizpour et al., 2015; Huh et al., 2016; Chu et al., 2016; Kornblith et al., 2019; Zhai et al., 2019; Locatello et al., 2020), e.g. model architecture, datasets and loss functions, on learned representations.

Table 2: Improving CLIP's transfer performance through simple datasets interventions. (*left*) Applying a simple bag-of-words classifier to identify data subsets with "high quality" captions (results with "descriptive" BLIP captions shown for comparison). (*right*) Using in-context learning with GPT-J to obtain (five) diverse captions for images (via paraphrasing) and then training $CLIP_S$.

| Dataset | Method | Preproc. | $\mu_{Tx}$ |
|---------|--------|----------|-----------|
| | SimCLR | - | $55.9 \pm 0.3$ |
| CC (100K) | CLIP | - | $53.1 \pm 0.2$ |
| | CLIP | Filter | $54.2 \pm 0.2$ |
| | SimCLR | - | $55.4 \pm 0.2$ |
| YFCC (500K) | CLIP | - | $58.8 \pm 0.2$ |
| | CLIP | BLIP | $61.8 \pm 0.2$ |
| | CLIP | Filter | $60.4 \pm 0.2$ |

| Dataset | Method | Caption | $\mu_{Tx}$ |
|---------|--------|---------|-----------|
| | SimCLR | - | $56.0 \pm 0.2$ |
| COCO (120K) | CLIP | Human | $57.5 \pm 0.1$ |
| | $CLIP_S$ | Human | $61.3 \pm 0.2$ |
| | $CLIP_S$ | GPT-J | $58.9 \pm 0.3$ |
| | SimCLR | - | $55.3 \pm 0.2$ |
| CC (200K) | CLIP | Human | $57.0 \pm 0.3$ |
| | $CLIP_S$ | GPT-J | $58.8 \pm 0.3$ |

**The utility of language in vision.** There is a long line of work on leveraging language to improve vision models (Quattoni et al., 2007; Srivastava & Salakhutdinov, 2012; Frome et al., 2013; Baltrušaitis et al., 2018; Guo et al., 2019). Recent studies have sought to investigate how integral language is to the performance of such multi-modal models. Fang et al. (2022) study a different property of CLIP—zero-shot robustness, rather than transfer learning—and show that it is comparable to that of a supervised classifier trained on the same YFCC images. Therefore, they conclude that data distribution is more important than language supervision. In concurrent work, (Nguyen et al., 2022) study the sensitivity of CLIP's zero-shot robustness to the pre-training dataset. However, unlike our work, they do not: (i) contrast CLIP against image-only methods trained on the same corpora, and (ii) attempt to explain what properties of the data are responsible for CLIP's sensitivity. Ruan et al. (2022) argue theoretically that the robustness of linear probes on CLIP's representations stems from pretraining with a large and diverse set of images and domain-agnostic augmentations $T(x)$. Most similar to our work are the studies by (Desai & Johnson, 2021) and Devillers et al. (2021), which study the role of language supervision on transfer performance in the context of VirTex (a CLIP precursor) and CLIP respectively. Notably, the two works draw opposite conclusions as to the utility of language compared to purely image-based approaches. This difference stems from the fact that neither of the works attempt to directly control for algorithmic, architectural, and data-related confounders. Our work performs a substantially more controlled study on the effect of language supervision, allowing us to make more direct claims than these works.

## 6 DISCUSSION

Our work takes a step towards resolving the debate as to whether multi-modality, and language in particular, can improve visual representation learning. A comparison of CLIP with a matched image-only SimCLR model reveals that neither form of supervision (using images alone or coupled with language) is strictly better than the other. Indeed, there are practical regimes where CLIP's performance cannot be matched using SimCLR with *any* amount of image data and others where language supervision is harmful. This is a direct consequence of CLIP's sensitivity to its pre-training data, especially its scale, descriptiveness, and variability of the captions. Through our analysis, we also discovered algorithmic improvements ($CLIP_S$) and dataset modifications (filtering and augmenting captions) to better take advantage of language supervision.

**Limitations.** Our exploration allows us to quantify the utility of language supervision (over using images alone) in a specific setting: transfer learning via probing on certain object recognition tasks (Kornblith et al., 2019). We view expanding the scope of our analysis as a direction for future work. Further, despite the significant steps we took to control the differences between CLIP and SimCLR, there are still some inconsistencies that have not been accounted for (discussed in Section 2). Nevertheless, the differences between our and previous results (e.g, Desai & Johnson, 2021; Devillers et al., 2021) suggest that we successfully pinned down some crucial confounders (architecture, augmentations, stochasticity, datasets, hyperparameters).

## ETHICS STATEMENT

Below, we discuss certain ethical concerns pertaining to our work:

- Although we rely on existing open source vision/multi-modal datasets for our analysis, prior work has raised concerns about some of these (or other similarly-sourced ones) being biased (Stock & Cisse, 2017; Yang et al., 2020; Birhane et al., 2021; Paullada et al., 2021) and violating privacy (Prabhu & Birhane, 2020; Yang et al., 2022).
- Our focus is on understanding the extent to which CLIP's representations are influenced by what the captions they are trained on describe. However, we sidestep whether or not this is always desirable. After all, recent studies (Birhane et al., 2021) show that vision-linguistic datasets have various biases and stereotypes, which we might not want our models to learn.
- In Section 4, we use large language models (in particular, GPT-J) to augment dataset captions via in-context learning. These models however are known to have their own limitations that might percolate into the generated captions.

## REPRODUCIBILITY STATEMENT

**Datasets:** All the pre-training/transfer learning datasets we use are open-source. In the supplementary material, we include certain variants of the COCO/CC/YFCC datasets we created as CSV files: namely synthetic COCO captions, filtered CC/YFCC samples, and GPT-J paraphrased captions.

**Code and hyperparameters:** We discuss implementation details including hyperparameter settings in Appendix A. We also include the code for training models in the supplementary material.

## ACKNOWLEDGEMENTS

We are grateful to Niladri Chatterji, Elisa Kreiss, Nimit Sohoni and Dimitris Tsipras for helpful discussions. SS is supported by Open Philanthropy, YD by a Knights-Hennessy Scholarship, and RT by the NSF GRFP under Grant No. DGE 1656518. We also thank Stanford HAI for a Google Cloud credits grant.

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
