# OpenReview forum: "Is a Caption Worth a Thousand Images? A Study on Representation Learning"
_ICLR.cc/2023/Conference — ICLR 2023 poster_

### Official Review · Reviewer_G3mZ · 2022-10-22

**Confidence:** 4
**Correctness:** 4
**Technical Novelty And Significance:** 4
**Empirical Novelty And Significance:** 4
**Recommendation:** 8

**Clarity, Quality, Novelty And Reproducibility:**

The paper is clearly written and complete with references and experimental details. In terms of quality, as stated above, the experiments are well structured, and do address several open questions (with previously contradicting results) in a clear, concise manner. The ideas of using SimCLR, and BLIP based captioning to study the various effects of text supervision are novel and provide insight into the behavior of such models. The results are reproducible with the provided code.

**Strength And Weaknesses:**

**Strengths**
1. The paper is well motivated. It is of great importance to study the exact effects of vision-language training versus pure image based training. The idea of using SimCLR as a controlled baseline is inspired.
2. The experiments have clear hypotheses and support the conclusions presented. Specifically, the use of multiple captions per image as text augmentations shows improvements over standard CLIP training, highlighting the importance of both scale and the variability of captions.
3. The presented approaches to improve VL training for more transferable representations are simple and intuitive.

**Weaknesses/Queries**
1. The overall transfer accuracy reported for SimCLR is much lower than that reported in Chen et al's original work. The difference primarily seems to be an effect of the training dataset chosen (Imagenet in the original v/s COCO here). Could the authors comment on this? One suggestion could be to use Imagenet and Imagenet-captions (Fang et al.) to better understand the difference.
2. Another improvement would be to study the effect of scaling up the architectures with more parameters (Resnet-2x, Resnet-4x) and perhaps with Transformer based architectures to understand if the conclusions hold across a variety of architectures/parameters.

However, I understand that these studies are time-consuming, and the paper in its current form is a valuable study in itself.


**Summary Of The Paper:**

This paper studies the transfer learning performance of vision language models (CLIP) in comparison with SimCLR. The primary motivation here is that SimCLR is essentially an image-image analogue of CLIP allowing for fair comparisons in performance while controlling for a variety of confounders. The paper claims three primary advantages of CLIP over VL. In terms of scaling, CLIP learns more transferrable representations when the data size increases. However, the required captions need to be descriptive. The authors leverage BLIP to score caption quality and show that datasets with higher caption quality lead to CLIP outperforming SimCLR as well as models trained with more low quality captions. The thirdly, the paper also provides experimental evidence that while variability in captions adversely affects VL training, this can be somewhat compensated by adding more data through text augmentations.

**Summary Of The Review:**

Overall, this paper is a positive contribution the community with a well structured study comparing the effect of language supervision to that of image based contrastive training. The authors control for several of the confounders that previous studies lack, and provide actionable insights that will be useful for further understanding vision-language models.

---

> ### Author Response · Authors · 2022-11-15
> **Author response**
>
> We thank the reviewer for taking the time to review our paper and for their kind feedback. We address individual comments below:
>
> >> The overall transfer accuracy reported for SimCLR is much lower than that reported in Chen et al's original work. The difference primarily seems to be an effect of the training dataset chosen (Imagenet in the original v/s COCO here). Could the authors comment on this? One suggestion could be to use Imagenet and Imagenet-captions (Fang et al.) to better understand the difference. https://github.com/mlfoundations/imagenet-captions
>
> We believe that a major contributor to the difference between our results and those in Table B.5 (Chen et al) is the additional number of training epochs (1000 in their case instead of 200 in ours for both CLIP and SimCLR). Note that Chen et al. also report performance (in-distribution) for 200 epoch pre-training and it is significantly lower than their 1000 epoch model. We expect the performance of both CLIP and SimCLR to go up as we train longer. That being said, the difference in image distributions (Imagenet vs COCO) could also contribute to this difference. We would be happy to add results from ImageNet in our future revision, although we note that Fang et al. provide captions (either Flickr title/tag/description) for only ~450K ImageNet images, with Flickr descriptions for only ~220K of these.
>
>
> >> Another improvement would be to study the effect of scaling up the architectures with more parameters (Resnet-2x, Resnet-4x) and perhaps with Transformer based architectures to understand if the conclusions hold across a variety of architectures/parameters.
>
> We agree with the reviewer that this would be an interesting direction for future research!

---

> > ### Comment · Reviewer_G3mZ · 2022-12-04
> > **Reponse**
> >
> > I thank the reviewers for their response. After considering the other reviewers and the responses as well as this one, I still believe this paper is a good attempt at explaining the robustness of vision language models. I will therefore stick with my original rating.

---

### Official Review · Reviewer_fwXk · 2022-10-23

**Confidence:** 4
**Correctness:** 3
**Technical Novelty And Significance:** 3
**Empirical Novelty And Significance:** 2
**Recommendation:** 5

**Clarity, Quality, Novelty And Reproducibility:**

The problem raised in this paper, which to explore the influence of setting and scale of vision-language dataset on the representation learning, is that somewhat novel and interesting.

The paper is clear and easy to read.

The experiment is shown to be reproduced easily.

**Strength And Weaknesses:**

Strength:
Raise an interesting topic that how many and what kind of captions are helpful for the vision-language pretraining.

Give out some ideas of what kind of a vision-language dataset may help model to learn better representations.

Sufficient experiments on different kinds of setting and scale of image-caption training sets to show the influence of different aspects of captions on the representation learning.

Weakness:

Some phenomena can be further explained for the reader to get better understanding. For example,
Why cannot the “more descriptive” BLIP-generated captions beat the performance of COCO dataset?

The paper evaluated the performance of representation learning only with classification-based transfer performance, while some other tasks can also be used to evaluate the capability of representations, i.e. image-text matching, and image textual grounding. The experiments from multiple different kinds of tasks can better support your claim.

Some experiments can be more completed. i.e. CLIP_s 2 and 10 in the left table of Fig. 5.

The proposed data augmentation methods are based on BLIP model which is also a supervised trained with large scale datasets. I doubt if the learned knowledge will be leaked during the data augmentation process especially in the image captioning process mentioned in Section 3.3. This can be view as a kind of knowledge distillation of BLIP model which makes it less supportive for the claim.

**Summary Of The Paper:**

In this paper, the author investigates different popular datasets to explore the influence of the style, and scale of a dataset on vision-language representation pretraining task. The authors find that a dataset with sufficiently large and descriptive captions will be helpful in the vision-language representation learning and the representation learning can be negatively influenced by variability of captions. According to the findings, the author proposed data augmentation methods based on BLIP to boost the pretraining performance of the methods trained with different variants of datasets.

**Summary Of The Review:**

In my view, this paper investigates an interesting topic and provide some preliminary rules of dataset design with some quantitative experiments for the vision-language learning.
However, some of experiments are not so convincing and some of them need to be completed to support the claim better, like more differentiation downstream tasks.
Therefore, I currently consider it marginally below the acceptance threshold.

---

> ### Author Response · Authors · 2022-11-15
> **Author response**
>
> We thank the reviewer for their feedback, and their kind comments regarding the topic we study and the detailedness of our experiments. We address specific concerns below:
>
> >> Some phenomena can be further explained for the reader to get better understanding. For example, Why cannot the “more descriptive” BLIP-generated captions beat the performance of COCO dataset?
>
> We see that the performance of CLIP trained with 100K samples from CC/YFCC labeled with BLIP is indeed comparable to CLIP trained with 100K COCO samples (within confidence intervals). In particular, compare CLIP_S with 5 captions in Figure 5 (left) to the 100K samples point for COCO in Appendix Figure 11.
>
> >> The paper evaluated the performance of representation learning only with classification-based transfer performance, while some other tasks can also be used to evaluate the capability of representations, i.e. image-text matching, and image textual grounding. The experiments from multiple different kinds of tasks can better support your claim.
>
> We would like to clarify that our goal is to understand whether the added language supervision in CLIP is key to its performance on downstream *vision* tasks. Building vision representations that transfer well has been a long-standing goal in the vision and ML communities (see discussion in [1,2]). In particular, prior works have shown that pre-trained vision representations tend to be more sample efficient [3,4], computationally efficient [5,6] and robust [7,8] compared to training directly on downstream tasks. Consequently, there have been several efforts to build improved pre-training methods for visual representation learning (e.g., all the work in self-supervised learning). One of the major appeals of CLIP was that it seemingly outperformed all prior approaches in this setting. This is precisely what motivated us to investigate CLIP and whether language was key to its remarkable performance. To evaluate the quality of vision representations, we focus on downstream classification tasks akin to the CLIP paper and prior work on evaluating visual representation quality (e.g., Kornblith et al.).
>
> >>  Some experiments can be more completed. i.e. CLIP_s 2 and 10 in the left table of Fig. 5.
>
> Due to computational constraints (generating BLIP captions is compute intensive), we performed more fine-grained experiments on the effect of the number of captions on CLIP_S’s performance on one dataset (YFCC). We updated the paper to include results on CC 100K with 2 captions.
>
> >> The proposed data augmentation methods are based on BLIP model which is also a supervised trained with large scale datasets. I doubt if the learned knowledge will be leaked during the data augmentation process especially in the image captioning process mentioned in Section 3.3. This can be view as a kind of knowledge distillation of BLIP model which makes it less supportive for the claim.
>
> We would like to clarify we are not suggesting BLIP captions as a proposed data augmentation scheme to train CLIP_S. In particular,
>
> - The goal of our analysis in Section 3.3 (“how many captions are enough?”) is to inform data collection: i.e., whether one should collect more images (and use CLIP) or more high-quality captions per image (and then use CLIP_S). Since at this moment, CC and YFCC only have a single caption available, we approximate the manual caption annotation we might perform in the future using BLIP-generated captions. Then, by training CLIP_S models with these BLIP captions, our goal is to approximate the relative performance gains from additional images vs more captions per image. Note that human captions are likely going to be strictly better than BLIP captions, so the gains from CLIP_S can only go further up.
>
> - In Section 4, we propose an automated data (caption) augmentation scheme for existing datasets, without collecting new captions. This procedure relies on large language models (GPT-J) which have been trained on language exclusively. Here, we rely on the in-context learning capabilities of these models to generate varied paraphrases of existing captions. In this case, since the model was never trained on image-caption pairs, there is no scope for knowledge distillation. As seen from Table 2 (right), training CLIP_S on GPT-J augmented captions does offer performance improvements. Using more powerful language models (e.g., non open-source models like GPT-3) could lead to further improvements.

---

### Official Review · Reviewer_Bcho · 2022-10-24

**Confidence:** 4
**Correctness:** 3
**Technical Novelty And Significance:** 2
**Empirical Novelty And Significance:** 3
**Recommendation:** 6

**Clarity, Quality, Novelty And Reproducibility:**

The paper is well-written and easy to follow. It is unclear which specific BLIP checkpoints they use for measuring the descriptiveness of captions and image captioning.

The paper mainly presents some novel empirical findings and two practical ways of improving CLIP.

The code has been submitted and the hyper-parameters have been well-documented.

**Strength And Weaknesses:**

Strengths:
1. The paper performs a relatively systematic study on how image-text contrastive loss is compared with an image-only contrastive loss and what properties of the training data can affect the performance of image-text contrastive loss. By drawing insights from their experiments, they can further demonstrate improvements.
2. When comparing image-text contrastive and image-only contrastive losses, they do a good job of identifying potential confounders and carefully control them as much as they can, making their conclusions more convincing compared with several previous papers.
3. Whether and to what extent language supervision can improve visual representation learning is an interesting and important topic. The conclusions of this paper can be helpful and inspire researchers in the future.

Weaknesses:
1. While image-text and image-only contrastive losses have some similarities, they are not mutually exclusive but can in fact be combined together [1]. Therefore, it would be better to see what are the fine-grained differences between the two paradigms and if combing them together can combine the best of them, instead of merely showing one is better than the other.
2. They find that the "descriptiveness" of caption data can affect the model performance and define the term "descriptiveness" as the extent to which they refer to what is contained in an image, which is somewhat vague and it seems that "descriptiveness" is an antonym of the commonly used term "noise". They use a retrieval system to quantify the "descriptiveness" of a caption and find that COCO>CC>YFCC, which seems like they are just measuring the quality of the datasets. Because CC and YFCC are scraped from the web and the image-caption pairs are noisy and not well-aligned, it is normal and well-known that noisy data can lead to bad performance. A more precise definition and a more suitable quantification method are required so that the difference between their conclusion and the well-known fact that "noisy data can be harmful" is more clear.
3. They choose to study the effect of caption variability on COCO, where the human-written captions are not quite diverse as noted in the paper. The Visual Genome dataset [2] can be used because it contains many more captions per image and each caption focuses on a specific part of its image, which can make their conclusions more convincing.



[1] Mu, Norman, Alexander Kirillov, David Wagner, and Saining Xie. "Slip: Self-supervision meets language-image pre-training." ECCV 2022.
[2] Krishna, R., Zhu, Y., Groth, O., Johnson, J., Hata, K., Kravitz, J., Chen, S., Kalantidis, Y., Li, L.J., Shamma, D.A. and Bernstein, M.S., 2017. Visual genome: Connecting language and vision using crowdsourced dense image annotations. IJCV.

**Summary Of The Paper:**

The paper performs a thorough investigation of CLIP-like image representation learning.

First, they try to compare the CLIP-like image-language contrastive loss and SimCLR-like image-only contrastive loss in a setting where many confounders are controlled. They conclude that adding language supervision can enable the model to learn more transferable representations than their image-only self-supervision method.

Then, they investigate the effect of data properties on CLIP by training CLIP on three different datasets. They find that language supervision can hurt the model transfer performance in low-data regimes but can be helpful in large-scale settings. Also, "descriptive" captions and low variability of captions can improve the model transfer performance.

Finally, based on their findings, they propose to filter non-descriptive captions by training a classifier to detect if a caption is COCO-like and mitigate caption variability by using GPT-J and COCO captions to generate new captions for existing image-caption datasets. They demonstrate the effectiveness of their proposed approach to CLIP.



**Summary Of The Review:**

The paper focuses on a specific question (i.e. does language supervision lead to more transferable representations than using images alone?) and presents a comprehensive analysis to answer this question. While many of their conclusions are interesting and insightful, I do find some issues that can be improved as stated in the weakness part. I am willing to discuss and change my score.

---

> ### Author Response · Authors · 2022-11-15
> **Author response (1/3)**
>
> We thank the reviewer for their detailed feedback, and for their kind comments regarding the thoroughness and relevance of our study. We address specific comments of the reviewer below,
> and have also revised our manuscript to do so.
>
> **Combining CLIP and SimCLR**
> >> While image-text and image-only contrastive losses have some similarities, they are not mutually exclusive but can in fact be combined together [1]. Therefore, it would be better to see what are the fine-grained differences between the two paradigms and if combing them together can combine the best of them, instead of merely showing one is better than the other.
>
> Our study was motivated by the rising popularity of CLIP as a backbone for downstream vision tasks, and was geared towards dissecting whether its success is tied to the added language supervision. We believe that precisely characterizing when and how language helps in CLIP is an important and thus far unaddressed question that could guide future development of joint vision-language models. Since our goal was to isolate the effect of language, we focused on designing a carefully matched image-only counterpart of CLIP contrasting the two methods across different pre-training settings. Thus, while we agree with the reviewer that combining the two methods could perhaps allow us to get the best of both worlds, we believe that this is outside the scope of our analysis and warrants an in-depth study in its own right.
>
> That being said, we conducted preliminary experiments on COCO where we incorporated the SimCLR loss into CLIP training. We present the results of this method (denoted as `CLIP + SimCLR loss`) alongside results using SimCLR/CLIP in isolation (from Table 1 in our paper) and added these to Appendix Table 6 in the revision. We see that on the majority of the tasks, combining the two methods outperforms them individually. There is likely significant room for improvement in these results by carefully tuning how the two losses are combined.
>
> | Model      | Aircraft | Birds |  Caltech101 | Caltech256 | Cars | CIFAR10 | CIFAR100 | DTD | Flowers | Food101 | Pets | SUN397 |
> | ----------- | ----------- | ----------- | ----------- | ----------- | ----------- | ----------- | ----------- | ----------- | ----------- | ----------- | ----------- | ----------- |
> | SimCLR |  40.6  | 18.5  | 71.5  | 58.6  | 31.5  | 82.1 |  57.3  | 61.7  | 77.4  | 58.7  | 57.3  | 51.9  |
> | CLIP |    41.4 | 17.6 | 73.2 | 60.4 | 35.8 | 83.6 | 60.8 | 65.7 | 80.5 | 60.9 | 57.0 | 50.8  |
> | CLIP + SimCLR loss      | 40.4  | 17.8	|  81.5 |	61.6	| 36.3 | 84.3 | 62.1 | 67.1 | 79.2 | 62.0 | 58.1 | 53.2 |

---

> ### Author Response · Authors · 2022-11-15
> **Author response (2/3)**
>
> **Descriptiveness vs noise**
>
> >> They find that the "descriptiveness" of caption data can affect the model performance and define the term "descriptiveness" as the extent to which they refer to what is contained in an image, which is somewhat vague and it seems that "descriptiveness" is an antonym of the commonly used term "noise". They use a retrieval system to quantify the "descriptiveness" of a caption and find that COCO>CC>YFCC, which seems like they are just measuring the quality of the datasets. Because CC and YFCC are scraped from the web and the image-caption pairs are noisy and not well-aligned, it is normal and well-known that noisy data can lead to bad performance. A more precise definition and a more suitable quantification method are required so that the difference between their conclusion and the well-known fact that "noisy data can be harmful" is more clear.
>
> We agree with the reviewer that we can be more precise in how we define descriptiveness, and have updated the manuscript to do so. As we state in the paper, this is a term that has been discussed in prior work on linguistics and accessibility. For instance, Bernardi et al. [1] define an image description as texts that “verbalize what can be seen in the image, i.e., they refer to the objects, actions, and attributes depicted, mention the scene type, etc.”. In contrast, typical image captions “provides personal, cultural, or historical context for the image.” [2].
>
> We would like to emphasize that “descriptiveness” is not an antonym to “noise”, and in fact there is a spectrum of *relevant* image captions that might not necessarily describe visually salient objects/attributes of an image.
>
> Alikhani et al. [3] provide an excellent taxonomy of relevant, non-noisy captions: Visible (“presents information that is intended to recognizably characterize what is depicted in the image”), Subjective (“describes the speaker’s reaction to, or evaluation of, what is depicted in the image”), Action (“describes an extended, dynamic process of which the moment captured in the image is a representative snapshot”), Story (“providing a free-standing description of the circumstances depicted in the image”) and Meta (“allows the reader to draw inferences not just about the scene depicted in the image but about the production and presentation of the image itself”). The "noise" category of captions described by the reviewer would be any caption that does not fall into those categories.
>
> Moreover, Alikhani et al. recruit expert annotators to categorize CC images based on this taxonomy. It turns out that CC captions are actually well-aligned with the corresponding images: only ~3% of the captions are irrelevant noise. The remainder of the captions are relevant, but might not always fall into the “Visible” category (their analogue to our notion of “descriptive”). To demonstrate that the same is true for YFCC, we manually annotate random dataset samples in Appendix Figure 12 using the taxonomy of Alikhani et al.. We also added this discussion to Appendix B.1.

---

> ### Author Response · Authors · 2022-11-15
> **Author response (3/3)**
>
> **Visual Genome**
>
> >> They choose to study the effect of caption variability on COCO, where the human-written captions are not quite diverse as noted in the paper. The Visual Genome dataset [2] can be used because it contains many more captions per image and each caption focuses on a specific part of its image, which can make their conclusions more convincing.
>
> We thank the reviewer for this suggestion. We conduct additional experiments on the VisualGenome dataset and present the results below (and in Appendix Table 9). We construct captions from the available region descriptions in the VisualGenome dataset (see Appendix Figure 6 for examples). Concretely, for each image, we construct:
>
> - 10 captions by randomly subsampling *all* the available region descriptions and concatenating them.
>
> - 10 captions by randomly subsampling the available region descriptions in the *first quadrant alone* and concatenating them. We do so in order to further understand the effect of incompleteness (i.e., describing only a part of the image) in captions on CLIP's performance.
>
> Using the two sets of captions per image ("full" or "quadrant") we can now train CLIP (using a single caption per image) or CLIP_S (stochastically sampling one of the ten captions). We then measure the transfer performance of the models:
>
> | Model    | Caption  | Aircraft | Birds |  Caltech101 | Caltech256 | Cars | CIFAR10 | CIFAR100 | DTD | Flowers | Food101 | Pets | SUN397 |
> | ----------- |  ----------- |----------- | ----------- | ----------- | ----------- | ----------- | ----------- | ----------- | ----------- | ----------- | ----------- | ----------- | ----------- |
> | CLIP |  Quadrant | 29.9	| 11.3 | 64.0 | 46.3 | 21.0 | 78.6 | 53.5 | 56.4 | 61.4  | 47.8 | 39.4 | 39.9 |
> | CLIP_S |   Quadrant |  42.8 | 16.9 | 79.8 | 59.2 | 38.5 | 81.0 | 57.4 | 62.8 | 77.6 | 56.0 | 54.5 | 50.0  |
> | CLIP | Full | 42.1 | 15.9 | 75.9 | 57.9 | 36.2 | 82.5 | 59.4 | 62.2 | 76.7 | 56.4 | 52.3 | 48.8 |
> | CLIP_S   | Full  | 44.2 | 18.2 | 82.4 | 62.5 | 39.4 | 83.0 | 59.7 | 65.4 | 81.2 | 60.3 | 56.1 | 54.6 |
>
> In this setting as well, we observe that:
>
> - *Incompleteness of caption significantly hurts CLIP*: cf. CLIP trained on a single caption based on region descriptions for the whole image vs one quadrant (rows 1 & 3).
> - *Stochastically sampling multiple captions during CLIP training significantly boosts its performance*: in both cases (for `full` and `quadrant` captions).
>
> **BLIP checkpoint**
> >> It is unclear which specific BLIP checkpoints they use for measuring the descriptiveness of captions and image captioning.
>
> Thanks to the reviewer for pointing out this omission. We use the checkpoint from: 'https://storage.googleapis.com/sfr-vision-language-research/BLIP/models/model_base_caption_capfilt_large.pth'
>
> References:
>
> [1] Bernardi, Raffaella, et al. "Automatic description generation from images: A survey of models, datasets, and evaluation measures." Journal of Artificial Intelligence Research 55 (2016)
>
> [2] Panofsky, E. (1939). Studies in Iconology. Oxford University Press
>
> [3] Alikhani, Malihe, et al. "Clue: Cross-modal coherence modeling for caption generation." arXiv preprint arXiv:2005.00908 (2020).

---

> > ### Comment · Reviewer_Bcho · 2022-11-24
> > **Review Update**
> >
> > I appreciate the additional experiments and explanations provided by the authors. Most of my concerns have been addressed in the responses and I have updated my score.
> >
> > I do believe that if two methods can be combined together and achieve better performance than any of the individual methods, it is worth performing an investigation of the combination. I am glad that the authors provide preliminary results in the response and I hope there can be a more in-depth analysis in the revised version.

---

### Official Review · Reviewer_TB29 · 2022-10-31

**Confidence:** 4
**Correctness:** 3
**Technical Novelty And Significance:** 2
**Empirical Novelty And Significance:** 2
**Recommendation:** 3

**Clarity, Quality, Novelty And Reproducibility:**

- Section 3.2 lacks relevant citations from language and vision research where similar observations have been repeatedly made. Flickr captions are varied (I am also unable to relate the section to Grice, 1975, perhaps it should be to [6]).

- The experimental settings and details in the paper are insufficient - such as:
   - Section 3.3, it is not clear how the captions are designed and further:
   - How should one quantify consistency?
   - How should one measure completeness - is this based on ground truth COCO objects for a particular image?
   - Without these details, it just seems arbitrary.
- Section 3.3 on how many captions are enough - what is “(high-quality)” captions - are the captions always factual? Nucleus sampling doesn’t always generate factual captions. Perhaps the observations due to plateauing is due to irrelevant captions?

[6] The construction of social reality. Searl 1995


**Strength And Weaknesses:**

Strength:
The paper contains an interesting exposition between single modality model  v/s multimodal model and the results confirm that multimodal models are indeed better at representational transfer (in general).

Weakness:
- The claim in the paper about “transfer of representations” seems general (and tad too strong) however this is only evaluated over vision heavy transfer learning benchmarks. It is not clear if the behaviour would be similar for transfer learning on vision and language related benchmarks.
- The salient observations (relating to language - especially descriptiveness and variability) in the paper are perhaps fairly trivial considering the vast amount of similar work from the area of vision and language and semiotics [1, 2, 3, 4, 5 inter alia].
- In general, the paper lacks empirical rigour, the paper contains a list of experimental interventions, but none are clear (expanding as the next point).

[1] Automatic description generation from images: A survey of models, datasets, and evaluation measures. Bernardi et al. 2016

[2] On the use of human reference data for evaluating automatic image descriptions. van Miltenburg. 2020

[3] Ways of seeing. Berger. 2008

[4] Semiotics: the basics. Chandler. 2007

[5] Underspecification in Scene Description-to-Depiction Tasks. Hutchinson et al. 2022


**Summary Of The Paper:**

This paper presents a controlled empirical study that compares the capability of representational transfer between unsupervised image-only model (mostly SimCLR) and unsupervised vision and language models (mostly CLIP). The central question that the paper tries to answer is whether unsupervised vision-and-language models that exploit “language information” are richer than image only models.  Controls in the paper are related to experimental settings such as training data, training architectures, transformations. The primary observations emphasise that vision-and-language models are richer with caveats on the type of vision and language parallel data and scale.

**Summary Of The Review:**

The current draft of the paper lacks experimental rigour, for an empirical paper most details are currently hand-wavy. The transfer based claims are perhaps supported for vision only transfer learning benchmarks.

---

> ### Author Response · Authors · 2022-11-15
> **Author response (1/2)**
>
> We thank the reviewer for their feedback and address specific concerns below.
>
>
> **Our analysis concerns only vision tasks**
>
> >>  “transfer of representations” seems general (and tad too strong) however this is only evaluated over vision heavy transfer learning benchmarks. It is not clear if the behaviour would be similar for transfer learning on vision and language related benchmarks.
>
> >> The transfer based claims are perhaps supported for vision only transfer learning benchmarks.
>
> We completely agree that our claims only hold for vision representations. This was actually the goal of our paper: to study whether language supervision was useful for learning better visual representations for downstream object recognition tasks. This can be seen from the the first sentences of our abstract and conclusion:
>
> -  “The development of CLIP has sparked a debate on whether
> adding language supervision can yield *vision models* with more transferable representations than traditional image-only methods.  Our work studies this question… to learn representations that generalize to *downstream classification tasks*.”
>
> -  “Our work takes a step towards resolving the debate as to whether multi-modality, and language in particular, can improve *visual representation learning*.”
>
> We recognize that this was not reiterated sufficiently in our introduction and we thank the reviewer for pointing this out. We updated the manuscript with this feedback. We would also like to emphasize that the specific setting we focus on—better visual representations for object recognition tasks—is a long-standing focus of work in ML (see discussion in Kornblith et al. and Radford et al.). In fact, one of the central contributions of the CLIP paper was a remarkable performance improvement over prior image-only approaches in this setting. However, the numerous differences between CLIP and these prior approaches, such as the data scale and composition, and varied algorithmic optimizations, make it hard to conclude whether these gains were due to the added language supervision. This is precisely what motivated our study, causing us to investigate whether (and when) language helps for visual representation learning.
>
> **Descriptiveness and variability**
> >> The salient observations (relating to language - especially descriptiveness and variability) in the paper are perhaps fairly trivial considering the vast amount of similar work from the area of vision and language and semiotics [1, 2, 3, 4, 5 inter alia].
>
> We thank the reviewer for these pointers. We would like to clarify that our work does not claim to discover differences in captions (in descriptiveness in particular) between datasets, nor do we view it to be our central contribution. For instance, in Section 3.2 we explicitly discuss prior work from linguistics and accessibility that has drawn the distinction between “descriptions” and “captions” and attempted to qualitatively describe how existing datasets fit into this categorization. The papers the reviewer has pointed out are in the same vein, and we are happy to add them to our discussion.
>
> The main contribution of our paper is to provide insight into whether (and when) added caption supervision in CLIP-style models leads to improved downstream performance (on classification tasks). Specifically, with regards to caption properties, our contribution is to:
>
> (i) Suggest an automated and quantitative approach for measuring descriptiveness (using a state-of-the-art retrieval model) in various image-language datasets.
>
> (ii) Empirically demonstrate that CLIP’s transfer performance on vision tasks is tightly coupled to the descriptiveness and variability of captions in its pre-training data.
>
> This seems to be entirely disparate from the studies the reviewer points to, and to the best of our knowledge, has not been demonstrated before.

---

> ### Author Response · Authors · 2022-11-15
> **Author response (2/2)**
>
> **Clarifications on experimental setting**
>
> >> Section 3.3, it is not clear how the captions are designed and further:
> How should one quantify consistency?
> How should one measure completeness - is this based on ground truth COCO objects for a particular image?
>
> The design of captions for COCO is detailed in appendix A.6. and we provide examples in Appendix Figure 8. We construct synthetic captions using the existing multi-object image labels in COCO, and thus can explicitly design them to (not) be consistent and complete. In order to clarify our generation procedure further, we illustrate it with a concrete example.
>
> Say an image X_1 has multi-object labels [“plate”, “cup”, “cup”, “refrigerator”], while image X_2 has multi-object labels [“refrigerator”, “cup”, “potted plant”, “tennis racket”].
>
> A consistent and complete caption would describe _all_ the image objects using a single descriptor/object in random order. For instance:
>
> - Caption(X_1) = “An image of a plate, two cups, and a refrigerator”.
> - Caption(X_2) = “An image of a refrigerator, potted plant, tennis racket and cup”.
>
> On the other hand, consistent and incomplete captions would still use a single descriptor/object, but might (randomly) omit certain image objects. For instance,
>
> - Caption(X_1) = “An image of a plate and cup”.
> - Caption(X_2) = “An image of a refrigerator, potted plant and tennis racket”.
>
> Finally, an inconsistent caption would use multiple descriptors/object in the image. For instance, a cup might be described (randomly) as a “cup” or “glass”. We obtain these descriptors using a set of curated synonyms based on the WordNet hierarchy. In this case, we also randomly vary the template. Concretely, inconsistent and incomplete captions might look like:
>
> - Caption(X_1) = “An image of a plate and glass”.
> - Caption(X_2) = “A photo of a fridge, tennis racket and a cup”.
>
> We added this example to A.6, and further clarified the generation procedure.
>
> ##
>
> >> [Figure 5] Section 3.3 on how many captions are enough - what is “(high-quality)” captions - are the captions always factual? Nucleus sampling doesn’t always generate factual captions. Perhaps the observations due to plateauing is due to irrelevant captions?
>
> We thank the reviewer for the interesting question and suggestion. We believe that this plateau is likely due to a combination of factors: a decrease in relevance and diversity of captions, as we sample more captions per image from BLIP.
>
> That being said, we would like to emphasize that we view our analysis in Figure 5 as a first step towards understanding the relative trade-offs from collecting more images vs collecting more captions per image. Here, the BLIP model is only meant to serve as a proxy for the manual/automated caption sourcing that might be done in practice. Ultimately, the exact number of captions per image at which CLIP’s performance plateaus will depend entirely on the procedure via which these captions are collected. We clarified this in Section 3.3.

---

### Author Response · Authors · 2022-11-15
**Author response**

We thank all the reviewers for taking the time to review our paper and for their feedback. We address individual reviewer's comments below, and have updated our manuscript based on their suggestions.

---

### Decision · Program_Chairs · 2023-01-20

**Decision:**

Accept: poster

**Justification For Why Not Higher Score:**

The problem domain (vision and language) is extremely difficult to ensure so-called empirical rigour, thus yielding limited insights to the general ICLR audiences. A poster would be a good position for this work.

**Justification For Why Not Lower Score:**

Well, if the authors could improve their presentation and ground their claims better, it could leverage this work away from the "could be rejected" category.

If it is rejected, I wouldn't feel too wrong either.

**Metareview: Summary, Strengths And Weaknesses:**

This paper studies the transfer learning performance of vision language models (CLIP) in comparison with SimCLR. The primary motivation here is that SimCLR is essentially an image-image analogue of CLIP allowing for fair comparisons in performance while controlling for a variety of confounders.

The paper claims three primary advantages of CLIP over VL. In terms of scaling, CLIP learns more transferrable representations when the data size increases. However, the required captions need to be descriptive. The authors leverage BLIP to score caption quality and show that datasets with higher caption quality lead to CLIP outperforming SimCLR as well as models trained with more low quality captions. The thirdly, the paper also provides experimental evidence that while variability in captions adversely affects VL training, this can be somewhat compensated by adding more data through text augmentations.

+ This work is of good importance to study given the popularity of vision-language training currently. It is of timely manner to present the findings to ICLR audiences.

Concerns still linger, and I double-checked with the one with "reject" rating without hearing back from him/her. Given the authors extended response to this specific reviewer, I think partially they have addressed the concerns raised.

**Note From Pc:**

if the above contains the word "oral" or "spotlight" please see: "oral" presentation means -> notable-top-5% and "spotlight" means -> notable-top-25%. As stated in our emails, we are disassociating presentation type from AC recommendations

**Summary Of Ac-Reviewer Meeting:**

N/A